# Learning Structured Densities via Infinite Dimensional Exponential Families

**Siqi Sun**
TTI Chicago
siqi.sun@ttic.edu

**Mladen Kolar**
University of Chicago
mkolar@chicagobooth.edu

**Jinbo Xu**
TTI Chicago
jinbo.xu@gmail.com

## Abstract

Learning the structure of a probabilistic graphical models is a well studied problem in the machine learning community due to its importance in many applications. Current approaches are mainly focused on learning the structure under restrictive parametric assumptions, which limits the applicability of these methods. In this paper, we study the problem of estimating the structure of a probabilistic graphical model without assuming a particular parametric model. We consider probabilities that are members of an infinite dimensional exponential family [4], which is parametrized by a reproducing kernel Hilbert space (RKHS) $\mathcal{H}$ and its kernel $k$. One difficulty in learning nonparametric densities is the evaluation of the normalizing constant. In order to avoid this issue, our procedure minimizes the penalized score matching objective [10, 11]. We show how to efficiently minimize the proposed objective using existing group lasso solvers. Furthermore, we prove that our procedure recovers the graph structure with high-probability under mild conditions. Simulation studies illustrate ability of our procedure to recover the true graph structure without the knowledge of the data generating process.

## 1 Introduction

Undirected graphical models, or Markov random fields [13], have been extensively studied and applied in fields ranging from computational biology [15, 28], to natural language processing [16, 20] and computer vision [9, 17]. In an undirected graphical model, conditional independence assumptions underlying a probability distribution are encoded in the graph structure. Furthermore, the joint probability density function can be factorized according to the cliques of the graph [14]. One of the fundamental problems in the literature is learning the structure of a graphical model given an i.i.d. sample from an unknown distribution. A lot of work has been done under specific parametric assumptions on the unknown distribution. For example, in Gaussian Graphical Models the structure of the graph is encoded by the sparsity pattern of the precision matrix [6, 30]. Similarly, in the context of exponential family graphical models, where the node conditional distribution given all the other nodes is a member of an exponential family, the structure is described by the non-zero coefficients [29]. Most existing approaches to learn the structure of a high-dimensional undirected graphical model are based on minimizing a penalized loss objective, where the loss is usually a log-likelihood or a composite likelihood and the penalty induces sparsity on the resulting parameter vector [see, for example, 6, 12, 18, 22, 24, 29, 30]. In addition to sparsity inducing penalties, methods that use other structural constraints have been proposed. For example, since many real-world networks are scale-free [1], several algorithms are designed specifically to learn structure of such networks

[5, 19]. Graphs tend to have cluster structure and learning simultaneously the structure and cluster assignment has been investigated [2, 27].

In this paper, we focus on learning the structure of a pairwise graphical models without assuming a parametric class of models. The main challenge in estimating nonparametric graphical models is computation of the log normalizing constant. To get around this problem, we propose to use score matching [10, 11] as a divergence, instead of the usual KL divergence, as it does not require evaluation of the log partition function. The probability density function is estimated by minimizing the expected distance between the model score function and the data score function, where the score function is defined as gradient of the corresponding probability density functions. The advantage of this measure is that the normalization constant is canceled out when computing the distance. In order to learn the underlying graph structure, we assume that the logarithm of the density is additive in node-wise and edge-wise potentials and use a sparsity inducing penalty to select non-zero edge potentials. As we will prove later, our procedure will allow us to consistently estimate the underlying graph structure.

The rest of paper is organized as follows. We first introduce the notations, background and related work. Then we formulate our model, establish a representer theorem and present a group lasso algorithm to optimize the objective. Next we prove that our estimator is consistent by showing that it can recover the true graph with high probability given sufficient number of samples. Finally the results for simulated data are presented to demonstrate the correctness of our algorithm empirically.

## 1.1 Notations

Let $[n]$ denote the set $\{1, 2, \ldots, n\}$. For a vector $\theta = (\theta_1, \ldots, \theta_d)^T \in \mathbb{R}^d$, let $\|\theta\|_p = (\sum_{i \in [d]} |\theta_i|^p)^{\frac{1}{p}}$ denote its $l_p$ norm. Let column vector $vec(D)$ denote the vectorization of matrix $D$, $cat(a, b)$ denote the concatenation of two vectors $a$ and $b$, and $mat(a_1^T, \ldots, a_d^T)$ the matrix with rows given by $a_1^T, \ldots, a_d^T$. For $\chi \subseteq \mathbb{R}^d$, let $L^p(\chi, p_0)$ denote the space of function for which the $p$-th power of absolute value is $p_0$ integrable; and for $f \in L^p(\chi, p_0)$, let $\|f\|_{L^p(\chi, p_0)} = \|f\|_p = (\int_\chi |f|^p dx)^{\frac{1}{p}}$ denote its $L^p$ norm. Throughout the paper, we denote $\mathcal{H}$ (or $\mathcal{H}_i, \mathcal{H}_{ij}$) as Hilbert space and $\langle \cdot, \cdot \rangle_{\mathcal{H}}, \|\cdot\|_{\mathcal{H}}$ as corresponding inner product and norm.

For any operator $C : \mathcal{H}_1 \to \mathcal{H}_2$, we use $\|C\|$ to denote the usual operator norm, which is defined as

$$\|C\| = \inf\{a \geq 0 : \|Cf\|_{\mathcal{H}_2} \leq a\|f\|_{\mathcal{H}_1} \text{ for all } f \in \mathcal{H}_1\};$$

and $\|C\|_{HS}$ to denote its Hilbert-Schmidt norm, which is defined as

$$\|C\|_{HS}^2 = \sum_{i \in I} \|Ce_i\|_{\mathcal{H}_2}^2,$$

where $e_i$ is an orthonormal basis of $\mathcal{H}$ for an index set $I$. Also, we use $\mathcal{R}(C)$ to denote operator $C$'s range space. For any $f \in \mathcal{H}_1$ and $g \in \mathcal{H}_2$, let $f \otimes g$ denote their tensor product.

## 2 Background & Related Work

### 2.1 Learning graphical models in exponential families

Let $x = (x_1, x_2, ..., x_d)$ be a $d$-dimensional random vector from a multivariate Gaussian distribution. It is well known that the conditional independency of two variables given all the others is encoded in the zero pattern of its precision matrix $\Omega$, that is, $x_i$ and $x_j$ are conditionally independent given $x_{-ij}$ if and only if $\Omega_{ij} = 0$, where $x_{-ij}$ is the vector of $x$ without $x_i$ and $x_j$. A sparse estimate of $\Omega$ can be obtained by maximum-likelihood (joint selection) or pseudo-likelihood (neighborhood selection) optimization with an added $l_1$ penalty [6, 22, 30]. Given $n$ independent realizations of $x$ (rows of $X \in \mathbb{R}^{n \times d}$), the penalized maximum-likelihood estimate of the precision matrix can be obtained as

$$\hat{\Omega}_\lambda = \arg\min_{\Omega \succ 0} \text{tr}(\hat{S}\Omega) - \log \det \Omega + \lambda \|\Omega\|_1, \tag{1}$$

where $\hat{S} = n^{-1} X^T X$ and $\lambda$ controls the sparsity level of estimated graph.

The pseudo-likelihood method estimates the neighborhood of a node $a$ by the non-zeros of the solution to a regularized linear model

$$\hat{\theta}_s = \arg\min_{\theta} \frac{1}{n}\|X_s - X_{-s}\theta\|_2^2 + \lambda\|\theta\|_1. \tag{2}$$

The estimated neighborhood is then $\hat{N}(s) = \{a : \theta_{sa} \neq 0\}$.

Another way to specify a parametric graphical model is by assuming that each node-conditional distributions is a part of the exponential family [29]. Specifically, the conditional distribution of $x_s$ given $x_{-s}$ is assumed to be

$$P(x_s|x_{-s}) = \exp\left(\sum_{t \in N(s)} \theta_{st} x_s x_t + C(x_s) - D(x_{-s}, \theta)\right), \tag{3}$$

where $C$ is the base measure, $D$ is the log-normalization constant and $N(s)$ is the neighborhood a the node $s$. Similar to (2), the neighborhood of each node can be estimated by minimizing the negative log-likelihood with $l_1$ penalty on $\theta$. The optimization is tractable when the normalization constant $D$ can be easily computed based on the model assumption. For example, under Poisson graphical model assumptions for count data, the normalization constant is $-\exp(\sum_{t \in N(s)} \theta_{st} x_t)$. When using the neighborhood estimation, the graph can be estimated as the union of the neighborhoods of each node, which leads to consistent graph estimation [22, 29].

## 2.2 Generalized Exponential Family and RKHS

We say $\mathcal{H}$ is a RKHS associated with kernel $k : \chi \times \chi \to \mathbb{R}_+$ if and only if for each $x \in \chi$, the following two conditions are satisfied: (1) $k(\cdot, x) \in \mathcal{H}$ and (2) it has reproducing properties such that $f(x) = \langle f, k(\cdot, x)\rangle_{\mathcal{H}}$ for all $f(\cdot) \in \mathcal{H}$, where $k$ is a symmetric and positive semidefinite function. Denote the RKHS $\mathcal{H}$ with kernel $k$ as $\mathcal{H}(k)$.

For any $f \in \mathcal{H}(k)$, there exists a set of $x_i$ and $\alpha_i$, such that $f(\cdot) = \sum_{i=1}^{\infty} \alpha_i k(\cdot, x_i)$. Similarly for any $g \in \mathcal{H}(k), g(\cdot) = \sum_{j=1}^{\infty} \beta_j k(\cdot, y_j)$, the inner product of $f$ and $g$ is defined as $\langle f, g\rangle_{\mathcal{H}} = \sum_{i,j=1}^{\infty} \alpha_i \beta_j k(x_i, y_j)$. Therefore the norm of $f$ simply is $\|f\|_{\mathcal{H}} = \sqrt{\sum_{i,j} \alpha_i \alpha_j k(x_i, x_j)}$. The summation is guaranteed to be larger than or equal to zero because the kernel $k$ is positive semidefinite.

We consider the exponential family in infinite dimensions [4], where

$$\mathcal{P} = \{p_f(x) = e^{f(x) - A(f)} q_0(x), x \in \chi; f \in \mathcal{F}\}$$

and the function space $\mathcal{F}$ is defined as

$$\mathcal{F} = \{f \in \mathcal{H}(k) : A(f) = \log \int_{\chi} e^{f(x)} q_0(x) dx < \infty\},$$

where $q_0(x)$ is the base measure, $A(f)$ is a generalized normalization constant such that $p_f(x)$ is a valid probability density function, and $\mathcal{H}$ is a RKHS [3] associated with kernel $k$. To see it as a generalization of the exponential family, we show some examples that can generate useful finite dimension exponential families:

- Normal: $\chi = \mathbb{R}$, $k(x, y) = xy + x^2 y^2$
- Poisson: $\chi = \mathbb{N} \cup \{0\}$, $k(x, y) = xy$
- Exponential: $\chi = \mathbb{R}_+$, $k(x, y) = xy$.

For more detailed information, please refer to [4].

When learning structure of a graphical model, we will further impose structural conditions on $\mathcal{H}(k)$ in order ensure that $\mathcal{F}$ consists of additive functions.

## 2.3 Score Matching

Score matching is a convenient procedure that allows for estimating a probability density without computing the normalizing constant [10, 11]. It is based on minimizing Fisher divergence

$$J(p\|p_0) = \frac{1}{2} \int p(x) \left\| \frac{\partial \log p(x)}{\partial x} - \frac{\partial \log p_0(x)}{\partial x} \right\|_2^2 dx, \tag{4}$$

where $\frac{\partial \log p(x)}{\partial x} = (\frac{\partial \log p(x)}{\partial x_1}, \ldots, \frac{\partial \log p(x)}{\partial x_d})$ is the score function. Observe that for $p(x, \theta) = \frac{1}{Z(\theta)} q(x, \theta)$ the normalization constant $Z(\theta)$ cancels out in the gradient computation, which makes the divergence independent of $Z(\theta)$. Since the score matching objective involves the unknown oracle probability density function $p_0$, it is typically not computable. However, under some mild conditions which we will discuss in METHODS section, (4) can be rewritten as

$$J(p \| p_0) = \int p_0(x) \sum_{i \in [d]} \frac{1}{2} \left( \frac{\partial \log p(x)}{\partial x_i} \right)^2 + \frac{\partial^2 \log p(x)}{\partial x_i^2} dx. \tag{5}$$

After substituting the expectation with an empirical average, we get

$$\hat{J}(p \| p_0) = \frac{1}{n} \sum_{a \in [n]} \sum_{i \in [d]} \frac{1}{2} \left( \frac{\partial \log p(X_a)}{\partial x_i} \right)^2 + \frac{\partial^2 \log p(X_a)}{\partial x_i^2}. \tag{6}$$

Compared to maximum likelihood estimation, minimizing $\hat{J}(p \| p_0)$ is computationally tractable. While we will be able to estimate $p_0$ only up to a scale factor, this will be sufficient for the purpose of graph structure estimation.

## 3 Methods

### 3.1 Model Formulation and Assumptions

We assume that the true probability density function $p_0$ is in $\mathcal{P}$. Furthermore, for simplicity we assume that

$$\log p_0(x) = f(x) = \sum_{\substack{i \leq j \\ (i,j) \in S}} f_{0,ij}(x_i, x_j),$$

where $f_{0,ii}(x_i, x_i)$ is a node potential and $f_{0,ij}(x_i, x_j)$ is an edge potential. The set $S$ denotes the edge set of the graph. Extensions to models where potentials are defined over larger cliques are possible. We further assume that $f_{0,ij} \in \mathcal{H}_{ij}(k_{ij})$, where $\mathcal{H}_{ij}$ is a RKHS with kernel $k_{ij}$. To simplify the notation, we use $f_{0,ij}(x)$ or $k_{ij}(\cdot, x)$ to denote $f_{0,ij}(x_i, x_j)$ and $k_{ij}(\cdot, (x_i, x_j))$. If the context is clear, we drop the subscript for norm or inner product. Define

$$\mathcal{H}(S) = \{ f = \sum_{(i,j) \in S} f_{ij} | f_{ij} \in \mathcal{H}_{ij} \} \tag{7}$$

as a set of functions that decompose as sum of bivariate functions on edge set $S$. Note that $\mathcal{H}(S)$ is also (a subset of) a RKHS with the norm $\|f\|^2_{\mathcal{H}(S)} = \sum_{(i,j) \in S} \|f_{ij}\|^2_{\mathcal{H}_{ij}}$ and kernel $k = \sum_{(i,j) \in S} k_{ij}$.

Let $\Omega(f) = \|f\|_{\mathcal{H},1} = \sum_{i \leq j} \|f_{ij}\|_{\mathcal{H}_{ij}}$. For any edge set $S$ (not necessarily the true edge set), we denote $\Omega_S(f_S) = \sum_{s \in S} \|\hat{f}_s\|_{\mathcal{H}_s}$ as the norm $\Omega$ reduced to $S$. Similarly, denote its dual norm as $\Omega_S^*[f_S] = \max_{\Omega_S(g_S) \leq 1} \langle f_S, g_S \rangle$ [25].

Under the assumption that the unknown $f_0$ is additive, the loss function becomes

$$
\begin{aligned}
J(f) =& \frac{1}{2} \int p_0(x) \sum_{i \in [d]} \left( \frac{\partial f(x)}{\partial x_i} - \frac{\partial f_0(x)}{\partial x_i} \right)^2 dx \\
=& \frac{1}{2} \sum_{i \in [d]} \sum_{j,j' \in [d]} \langle f_{ij} - f_{0,ij}, \int p_0(x) \frac{\partial k_{ij}(\cdot, (x_i, x_j))}{\partial x_i} \otimes \frac{\partial k_{ij'}(\cdot, (x_i, x_{j'}))}{\partial x_i} dx (f_{ij'} - f_{0,ij'}) \rangle \\
=& \frac{1}{2} \sum_{i \in [d]} \sum_{j,j' \in [d]} \langle f_{ij} - f_{0,ij}, C_{ijij'}(f_{ij'} - f_{0,ij'}) \rangle.
\end{aligned}
$$

Intuitively, $C$ can be viewed as a $d^2$ matrix, and the operator at position $(ij, ij')$ is $C_{ij,ij'}$. For general $(ij, i'j'), i \neq i'$ the corresponding operator simply is 0. Define $C_{SS'}$ as

$$\int p_0(x) \sum_{(i,j) \in S, (i',j') \in S'} \frac{\partial k_{ij}(\cdot, (x_i, x_j))}{\partial x_i} \otimes \frac{\partial k_{i'j'}(\cdot, (x_{i'}, x_{j'}))}{\partial x_i} dx,$$

which intuitively can be treated as a sub matrix of $C$ with rows $S$ and columns $S'$. We will use this notation intensively in the main theorem and its proof.

Following [26], we make the following assumptions.

**A1.** Each $k_{ij}$ is twice differentiable on $\chi \times \chi$.

**A2.** For any $i$ and $\tilde{x}_j \in \chi_j = [a_j, b_j]$, we assume that

$$\lim_{x_i \to a_i^+ \text{ or } b_i^-} \frac{\partial^2 k_{ij}(x,y)}{\partial x_i \partial y_i} \mid_{y=x} p_0^2(x) = 0,$$

where $x = (x_i, \tilde{x}_j)$ and $a_i, b_i$ could be $-\infty$ or $\infty$.

**A3.** This condition ensures that $J(p\|p_0) < \infty$ for any $p \in \mathcal{P}$ [for more details see 26]:

$$\|\frac{\partial k_{ij}(\cdot, x)}{\partial x_i}\|_{\mathcal{H}_{ij}} \in L^2(\chi, p_0), \|\frac{\partial^2 k_{ij}(\cdot, x)}{\partial x_i^2}\|_{\mathcal{H}_{ij}} \in L^2(\chi, p_0).$$

**A4.** The operator $C_{SS}$, is compact and the smallest eigenvalue $\omega_{\min} = \lambda_{min}(C_{SS}) > 0$.

**A5.** $\Omega_{S^c}^*[C_{S^c S}C_{SS}^{-1}] \leq 1 - \eta$, where $\eta > 0$.

**A6.** $f_0 \in \mathcal{R}(C)$, which means there exists $\gamma \in \mathcal{H}$, such that $f_0 = C\gamma$. $f_0$ is the oracle function.

We will discuss the definition of operator $C$ and $\Omega^*$ in section 4. Compared with [29], A4 can be interpreted as the dependency condition and the A5 is the incoherence condition, which is a standard condition for structure learning in high dimensional statistical estimators.

## 3.2 Estimation Procedure

We estimate $f$ by minimizing the following penalized score matching objective

$$\min_f \hat{\mathcal{L}}_\mu(f) = \hat{J}(f) + \frac{\mu}{2}\|f\|_{\mathcal{H},1}$$
$$s.t. \quad f_{ij} \in \mathcal{H}_{ij}, \tag{8}$$

where $\hat{J}(f)$ is given in (6). The norm $\|f\|_{\mathcal{H},1} = \sum_{i \leq j} \|f_{ij}\|_{\mathcal{H}_{ij}}$ is used as a sparsity inducing penalty. A simplified form of $\hat{J}(f)$ is given below that will lead to efficient algorithm for solving (8).

The following theorem states that the score matching objective can be written as a penalized quadratic function on $f$.

**Theorem 3.1** *(i) The score matching objective can be represented as*

$$\mathcal{L}_\mu(f) = \frac{1}{2}\langle f - f_0, C(f - f_0)\rangle + \frac{\mu}{2}\|f\|_{\mathcal{H},1} \tag{9}$$

*where $C = \int p_0(x) \sum_{i \in [d]} \frac{\partial k(\cdot, x)}{\partial x_i} \otimes \frac{\partial k(\cdot, x)}{\partial x_i} dx$ is a trace operator.*

*(ii) Given observed data $X_{n \times d}$, the empirical estimation of $\mathcal{L}_\mu$ is*

$$\hat{\mathcal{L}}_\mu(f) = \frac{1}{2}\langle f, \hat{C}f\rangle + \sum_{i \leq j}\langle f_{ij}, -\hat{\xi}_{ij}\rangle + \frac{\mu}{2}\|f\|_{\mathcal{H},1} + const \tag{10}$$

*where $\hat{C} = \frac{1}{n}\sum_{a \in [n]}\sum_{i \in [d]} \frac{\partial k(\cdot, X_a)}{\partial x_i} \otimes \frac{\partial k(\cdot, X_a)}{\partial x_i}$ and $\hat{\xi}_{ij} = \frac{1}{n}\sum_{a \in [n]} \frac{\partial^2 k_{ij}(\cdot, (X_{ai}, X_{aj}))}{\partial x_i^2} + \frac{\partial^2 k_{ij}(\cdot, (X_{ai}, X_{aj}))}{\partial x_j^2}$ if $i \neq j$, or $\hat{\xi}_{ij} = \frac{1}{n}\sum_{a \in n} \frac{\partial^2 k_{ij}(\cdot, (X_{ai}, X_{aj}))}{\partial x_i^2}$ otherwise.*

Please refer to our supplementary material for detailed proof [1].

The above theorem still requires us to minimize over $\mathcal{F}$. Our next results shows that the solution is finite dimensional. That is, we establish a representer theorem for our problem.

**Theorem 3.2** *(i) The solution to* (10) *can be represented as*

$$\hat{f}_{ij}(\cdot) = \sum_{b\in[n]} \beta_{bij}\frac{\partial k_{ij}(\cdot,(X_{bi},X_{bj}))}{\partial x_i} + \beta_{bji}\frac{\partial k_{ij}(\cdot,(X_{bi},X_{bj}))}{\partial x_j} + \alpha_{ij}\hat{\xi}_{ij}, \tag{11}$$

*where $i \leq j$.*
*(ii) Minimizing* (10) *is equivalent to minimizing the following quadratic function:*

$$
\begin{aligned}
&\frac{1}{2n}\sum_{ai}\left(\sum_{bj}(\beta_{bij}G^{ab}_{ij11}+\beta_{bji}G^{ab}_{ij12})+\sum_j \alpha_{ij}h^{1a}_{ij}\right)^2 \\
&+\sum_{i\leq j}\sum_b(\beta_{bij}h^{1b}_{ij}+\beta_{bji}h^{2b}_{ij})+\sum_{i\leq j}\alpha_{ij}\|\hat{\xi}_{ij}\|^2+\frac{\mu}{2}\|f\|_{\mathcal{H},1} \\
&=\frac{1}{2n}\sum_{ai}(D^T_{ai}\cdot\theta)^2+E^t\theta+\frac{\mu}{2}\sum_{i\leq j}\sqrt{\theta^t_{ij}F_{ij}\theta_{ij}}
\end{aligned} \tag{12}
$$

*where $G^{ab}_{ijrs}=\frac{\partial^2 k_{ij}(X_a,X_b)}{\partial x_r \partial y_s}$, $h^{rb}_{ij}=\langle\frac{\partial k_{ij}(\cdot,X_b)}{\partial x_r},\hat{\xi}_{ij}\rangle$ are constant that only depends on X, $\theta = cat(vec(\alpha),vec(\beta))$ is the vector parameter and $\theta_{ij}=cat(\alpha_{ij},vec(\beta_{\cdot ij}))$ is a group of parameters. $D_{ai}, E$ and $F$ are corresponding constant vectors and matrices based on $G$, $h$ and the order of parameters. Then the above problem can be solved by group lasso [7, 21].*

The first part of theorem states our represener theorem, and the second part is obtained by plugging in (11) to (10). See supplementary material for a detailed proof. Theorem 3.2 provides us with an efficient way to minimize (8), as it reduced the optimization to a group lasso problem for which many efficient solvers exist.

Let $\hat{f}^\mu = \arg\min_{f\in\mathcal{H}}\hat{\mathcal{L}}_\mu(f)$ denote the solution to (12). We can estimate the graph as follows:

$$\hat{S}_\mu = \{(i,j):\|\hat{f}^\mu_{ij}\|\neq 0\}, \tag{13}$$

That is, the graph is encoded in the sparsity pattern of $\hat{f}^\mu$.

# 4 Statistical Guarantees

In this section we study statistical properties of the proposed estimator (13). Let $S$ denote the true edge set and $S^c$ its complement. We prove that $\hat{S}_\mu$ recovers $S$ with high probability when the sample size $n$ is sufficiently large.

Denote $D = mat(D^T_{11},\ldots,D^T_{ai},\ldots,D^T_{nd})$. We will need the following result on the estimated operator $\hat{C}$,

**Proposition 4.1 (Lemma 5 in [8] or Theorem 5 in [26] )** *(Properties of $C$)*

1. $\|\hat{C}-C\|_{HS}=O_{p_0}(n^{-\frac{1}{2}})$

2. $\|(C+\mu L)^{-1}\|\leq\frac{1}{\mu\min diag(L)}$, $\|C(C+\mu L)^{-1}\|\leq 1$, *where $\mu>0$ and $L$ is diagonal with positive constants.*

The following result gives first order optimality conditions for the optimization problem (8).

**Proposition 4.2** *(Optimality Condition)*
*$\hat{J}(f)+\frac{\mu}{2}\Omega(f)^2$ achieves optimality when the following two conditions are satisfied:*

$$(1)\quad \nabla_{f_s}\hat{J}(f)+\mu\Omega(f)\frac{f_s}{\|f_s\|_{\mathcal{H}_s}}=0\quad\forall s\in S$$

$$(2)\quad \Omega^*_{S^c}[\nabla_{f_{S^c}}\hat{J}(f)]\leq\mu\Omega(f).$$

With these preliminary results, we have the following main results.

**Theorem 4.3** *Assume that conditions* **A1**-**A7** *are satisfied. The regularization parameter $\mu$ is selected at the order of $n^{-\frac{1}{4}}$ and satisfies $\mu \leq \frac{\eta \kappa_{\min} \omega_{\min}}{4(1-\eta)\kappa_{\max}\sqrt{|S|}+\frac{\eta}{5}}$, where $\kappa_{\min} = \min_{s \in S} \|f_s^*\| > 0$ and $\kappa_{\max} = \max_{s \in S} \|f_s^*\| > 0$. Then $P(\hat{S}_\mu = S) \to 1$.*

**Proof Idea**: The theorem above is the main theoretical guarantee for our score matching estimator. We use the "witness" proof framework inspired by [23, 29]. Let $f^*$ denote the true density function and $p^*$ the probability density function. We first construct a solution $\hat{f}_S$ on true edge set $S$ as

$$\hat{f}_S = \min_{f_{S^c}=0} \hat{J}(f) + \frac{\mu}{2}\Big( \sum_{(i,j)\in S} \|f_{ij}\| \Big)^2 \tag{14}$$

and set $\hat{f}_{S^c}$ as zero. Using Proposition 4.1, we prove that $\|\hat{f}_S - f_S^*\| = O_p(n^{-\frac{1}{4}})$. Then we compute the subgradient on $S^c$ and prove that its dual norm is upper bounded by $\mu\Omega(f)$ by using assumptions A4, A5 and A6. Therefore we construct a solution that satisfied the optimality condition and converges in probability to the true graph. Refer to supplementary material for detailed proof.

## 5  Experiments

We illustrate performance of our method on two simulations. In our experiments, we use the same kernel defined as follows:

$$k(x,y) = \exp(-\frac{\|x-y\|_2^2}{2\sigma^2}) + r(x^T y + c)^2, \tag{15}$$

that is, the summation of a Gaussian kernel and a polynomial kernel. We set $\sigma^2 = 1.5, r = 0.1$ and $c = 0.5$ for all the simulations.

We report the true positive rate vs false positive rate (ROC) curve to measure the performance of different procedures. Let $S$ be the true edge set, and let $\hat{S}^\mu$ be the estimated graph. The true positive rate is defined as $\text{TPR}_\mu = \frac{|S=1 \text{ and } \hat{S}^\mu=1|}{|S=1|}$, and false positive rate is $\text{FPR}_\mu = \frac{|\hat{S}^\mu=1 \text{ and } S=0|}{|S=0|}$, where $|\cdot|$ is the cardinality of the set. The curve is then plotted based on 100 uniformly-sampled regularization parameters and based on 20 independent runs.

In the first simulation, we apply our algorithm to data sampled from a simple chain graph-based Gaussian model (see Figure 1 for detail), and compare its performance with glasso [6]. We use the same sampling method as in [31] to generate the data: we set $\Omega_s = 0.4$ for $s \in S$ and its diagonal to a constant such that $\Omega$ is positive definite. We set the dimension $d$ to 25 and change the sample size $n \in \{20, 40, 60, 80, 100\}$ data points.

Except for the low sample size case ($n = 20$), the performance of our method is comparable with glasso, without utilizing the fact that the underlying distribution is of a particular parametric form. Intuitively, to capture the graph structure, the proposed nonparametric method requires more data because of much weaker assumptions.

To further show the strength of our algorithm, we test it on a nonparanormal (NPN) distribution ([18]). A random vector $x = (x_1, \ldots, x_p)$ has a nonparanormal distribution if there exist functions $(f_1, \ldots, f_p)$ such that $(f_1(x_1), \ldots, f_d(x_d)) \sim N(\mu, \Sigma)$. When $f$ is monotone and differentiable, the probability density function is given by

$$P(x) = \frac{1}{(2\pi)^{\frac{p}{2}}|\Sigma|^{\frac{1}{2}}} \exp\{-\frac{1}{2}(f(x)-\mu)^T\Sigma^{-1}(f(x)-\mu)\} \prod_j |f_j'|.$$

Here the graph structure is still encoded in the sparsity pattern of $\Omega = \Sigma^{-1}$, that is, $x_i \perp x_j | x_{-i,j}$ if and only if $\Omega_{ij} = 0$ [18].

In our experiments we use the "Symmetric Power Transformation" [18], that is,

$$f_j(z_j) = \sigma_j\Big(\frac{g_0(z_j - \mu_j)}{\sqrt{\int g_0^2(t-\mu_j)\phi(\frac{t-\mu_j}{\sigma_j})dt}}\Big) + \mu_j,$$

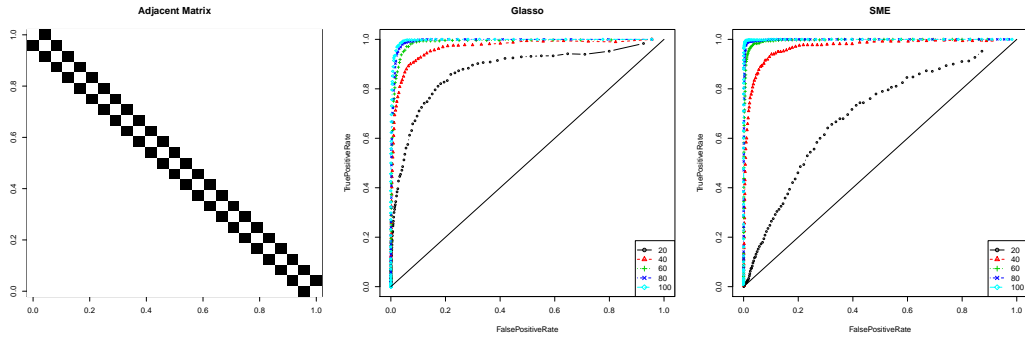

Figure 1: The estimation results for Gaussian graphical models. **left**: The adjacent matrix of true graph. **center**: the ROC curve of glasso. **right**: the ROC curve of score matching estimator (SME).

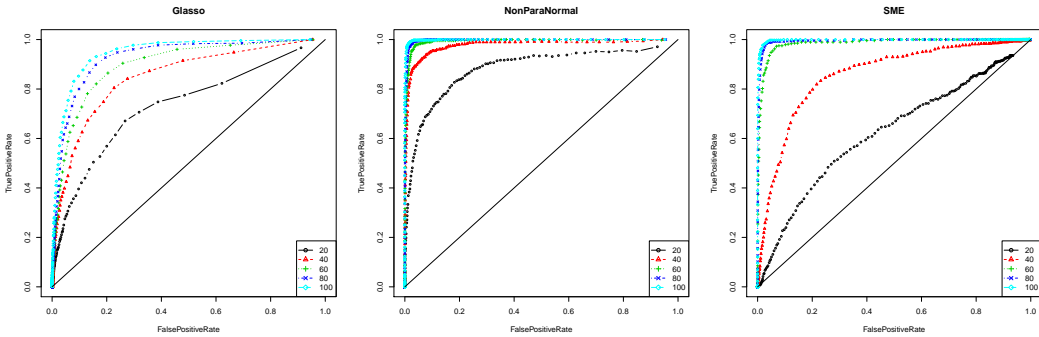

Figure 2: The estimated ROC curves of nonparanormal graphical models for glasso (left), NPN (center) and SME (right).

where $g_0(t) = \text{sign}(t)|t|^{\alpha}$, to transform data. For comparison with graph lasso, we first use a truncation method to Gaussianize the data, and then apply graphical lasso to the transformed data. See [18, 31] for details. From figure 2, without knowing the underlying data distribution, the score matching estimator outperforms glasso, and show similar results to nonparanormal when the sample size is large.

## 6   Discussion

In this paper, we have proposed a new procedure for learning the structure of a nonparametric graphical model. Our procedure is based on minimizing a penalized score matching objective, which can be performed efficiently using existing group lasso solvers. Particularly appealing aspect of our approach is that it does not require computing the normalization constant. Therefore, our procedure can be applied to a very broad family of infinite dimensional exponential families. We have established that the procedure provably recovers the true underlying graphical structure with high-probability under mild conditions. In the future, we plan to investigate more efficient algorithms for solving (10), since it is often the case that $\hat{C}$ is well structured and can be efficiently approximated.

#### Acknowledgments

The authors are grateful to the financial support from National Institutes of Health R01GM0897532, National Science Foundation CAREER award CCF-1149811 and IBM Corporation Faculty Research Fund at the University of Chicago Booth School of Business. This work was completed in part with resources provided by the University of Chicago Research Computing Center.

## Footnotes

[1]Please visit ttic.uchicago.edu/~siqi for supplementary material and code.

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
