[Reviews · NeurIPS 2015]

Submitted by Assigned_Reviewer_1

Summary:

The paper provide a new nonparametric model for estimating the structure of a probabilistic graphical models considering infinite dimensional exponential family (based on infinite dimensional reproducing kernel Hilbert space). Their estimation is based on score matching which avoids intractable/expensive computation of of the log partition. They show that the estimator can be solved using any current algorithm for solving group lasso. They also show the theoretical consistent analysis of the estimated sparse structure using primal-witness approach.

Qualitative Evaluation

Overall, not a well written paper. This paper does not provide a good review of other related tools based on score matching. Authors provide the theoretical analysis for the claims in the paper but they do not compare their results with existing approaches - a section on related work would have been helpful. The analysis/method in the paper is mainly based on [28] but authors do not discuss the relationship of their work with [28], they just cite this paper as they directly used some of their results without any comments.

It is not clear how good/bad is their consistency analysis against existing papers in the literature. Although not discussed in detail in the main paper, the non-asymptotic rates of convergence seem much worse than the parametric setting, or even the semi-parametric setting as in [18]. So, with finite samples, it is unclear when/why the method should be used. Again, there was no discussion in the paper to illuminate this or any other point.

The problem addressed in the paper is an interesting problem and a challenging problem. The model is based on score matching which is already addressed in [28], but they provide the connection with group lasso which provides a good solver for their model. However, currently algorithm only works with l1 norm and it is not clear if this model can be applied with other regularizers to capture different structure and if it can be done, how challenging is the solver. The consistency analysis is based on primal-dual witness which can be difficult to be extended to other regularizers.
Summary: The paper provide a new nonparametric model for estimating the structure of a probabilistic graphical models considering infinite dimensional exponential family. The paper is not well written - while the results look potentially interesting, no comparison is done with existing methods in terms of the technical results. It seems the non-asymptotic rates obtained are worse than certain existing results. The paper is mainly based on [28], so a more clear exposition on the novelty will be helpful.

Submitted by Assigned_Reviewer_2

This paper learns a nonparametric graphical model via score matching, which effectively bypasses the need to estimate the normalizing constant. Theory and simulations were provided.

Quality: The nonparametric graphical model and the neighborhood estimation method are not entirely new. However,

what is interesting is that the need for computing the normalizing constant is bypassed by using score matching. Parametrized by RKHS, the formulation in (8) is shown to achieve neighborhood selection consistency.

Clarity: The paper is well presented.

Originality and significance: The paper is original in that using score matching, one does not need to evaluate the normalizing constant.
Summary: The score matching objective proposed in the paper for learning nonparametric graphical models is free of normalizing constants. This is an interesting idea.

Submitted by Assigned_Reviewer_3

Summary: The paper presents a novel nonparametric procedure to estimate the structure of a sparse undirected probabilistic graphical model from data. It is based on using kernels to approximate the resulting distribution by an infinite dimensional member of the exponential family, and uses the RKHS representer theorem to reduce the corresponding optimization problem to one that can be efficiently tackled using existing group lasso solvers. It is well written, clearly explained, and utilises at times quite challenging maths to arrive at an elegant solution. The ideas and techniques developed in this paper should be of interest to many researchers in the field of structure learning. Only weakness is evaluation section, which should have included more complex graph structures with weaker edges to give a better indication of performance in practice.

Quality: Good idea, well executed. All steps in the reduction of the (penalised) score matching objective to the quadratic minimization problem make sense, although I did not check the detailed proofs in the supplementary material, and the abstract parameterisation in (12) is quite difficult to follow.

As stated above, only weakness is the evaluation: the (very sparse) chain graph used in the examples in combination with edge strength \Omega_s = 0.4 (line 359) makes for rather easy structures. As a result the performance in FIg.1 is already very good at N=60 data points to nearly perfect at N=100 ... something that is very a-typical for real-world structure inference problems. I would have preferred to see the method evaluated on more realistic models to get a better impression of the actual performance obtained by the method.

Clarity: Very good: problem + main idea is well described, concepts and formulas are carefully defined, and the paper does in good job in trying to make the overall argument accessible to most readers. For example, section 2.3 was compact but very helpful being less familiar with score matching. However, some of the steps in 3 could still do with a few hints to make it easier to follow.

Originality: The paper obviously builds on many existing results, but as far as I can tell the crucial contribution culminating in Theorem 3.2 is new, elegant, and hopefully very useful.

Significance: Significant addition to state-of-the-art methods in structure inference. I expect it to lead to various new ideas and applications.

Details: 53: check sentence 'we focus ... models,' 76: typo -> 'integrable' 98: typo -> 'realizations'

128: typo 'RHKS' -> RKHS' 202: rephrase 'an additive assumption on' 222: currently A.2 is formulated as an incomplete assumption

314: 'and D is diagonal' -> I did not get this from the definition in Theorem 3.2 358: for more realistic models sample \Omega_s from e.g. [0.05-0.8] 407: explain how exactly the NPN ROC-curve in Fig2b is obtained (what different objective is optimized in comparison to glasso and SME) 419: typo -> 'nonparametric' also: 'for learning the structure of a ..'
Summary: The paper presents a kernel based nonparametric procedure to efficiently estimate the structure of sparse undirected probabilistic graphical models from data using existing group lasso solvers.

It is highly relevant, well written, and utilises at times quite challenging maths to arrive at an elegant solution: clearly belongs in the conference.

Submitted by Assigned_Reviewer_4

I have read the authors' rebuttal and it did not change my opinion of the paper. -------- 1 Summary of the paper:

This paper proposes to learn the structure of a kernel-based graphical model encoding a joint probability distribution (probabilities belonging to an infinite dimensional exponential family[1]). By using score-matching as the objective of the learning procedure, the normalisation constant does not need to be estimated, which is scalable and sufficient to learn the structure. Moreover, because of the kernel-based parameterisation, the optimisation problem reduces to an instance of the group lasso. The structure recovered by the algorithm is asymptotically consistent. The method proposed is compared to the graphical lasso to estimate a gaussian distribution and to both glasso and an algorithm knowing the model family on a paranormal distribution

[1] S. Canu and A. Smola. Kernel methods and the exponential family. Neurocomputing, 69(7):714-720, 2006.

2 Summary of the review: While I did not verify the theoretical analysis provided in the paper and the experiments could be explained better, I find the paper well organized, the contributions reasonably clear, the idea very interesting and potentially impactful.

3 Qualitative evaluation

Quality: medium + extensive theoretical analysis (not checked fully, seems sound) +- small scale empirical evaluation - no discussion or empirical evaluation of how to choose the kernel parameters. Is a grid-search + cross-validation feasible and sufficient? How sensible is the result to the parameters chosen? - no code provided

Clarity: low + well organised paper, clear language +- lots of mathematical developments that are not easy to follow, especially with notations not necessarily introduced in the main paper.

- the parameters of the second model learned in the experiments are not provided.

Originality: good + I am not familiar with kernel-based methods or kernel-based graphical models, but I could not find really similar works beyond the ones cited.

+ significant development with respect to previously published cited work - still, other works using kernel with PGMs exists. It could be interesting to relate the paper under review to them, for the informed reader like myself. For example, a quick search on google yielded the following papers: "Nonparametric Tree Graphical Models via Kernel Embeddings", "Kernel Conditional Random Fields: Representation and Clique Selection", "Learning Graphical Models with Mercer Kernels".

Significance: good + structure learning for non-gaussian variable is an important open problem + I find the proposed idea very interesting

Impact: 2

4 Other Comments:

Details: . I understand you focus on the structure estimation problem, but can the normalisation constant be estimated easily enough when the model has been learned if one also wants the density? Are there some restricted families of kernel that make it possible? . I think section headers should not be all capitalized. . I am not familiar with RKHS. So I could not easily go from l204 to l207. I would suggest deriving it in the supplementary material. . a table of all notations in the supplementary material would make the paper a bit easier to follow. . l 319: why is it Omega(f)^2? (was Omega(f) in equation 8) . experiment 2: what is the dimensionality of the problem? What are the parameter values? . it would be nice to have the expression for D, E and F in theorem 3.2, for example in the appendix.

typos: . l25: evaluation of --> the evaluation of . l53: "the" structure (lots of similar missing "the" in the paper - though I'm not a native speaker.) . l 76: integrable . l86: H_2?

. l233: S^c not defined . l323: Omega*_{S^c} not defined
Summary: While I did not verify the theoretical analysis provided in the paper and the experiments could be explained better, I find the paper well organized, the contributions reasonably clear, the idea very interesting and potentially impactful.

Author Feedback
Author rebuttal: We thank the reviewers for their thoughtful comments that will lead to improved quality of the manuscript. We will revise our manuscript accordingly to make it more self-contained. We will also improve introduction and add more content on related work. Below, we address the major concerns.

Reviewer 3

Thank you for the additional references. There is a substantial body of work on learning densities in a nonparametric form. Our procedure is related to this line of research. Novel aspects of our approach are that we consider a particular form for the density and use score matching as a divergence. We will provide more extensive literature review in the revision.

Our computation procedure boils down to solving a group-lasso-like objective. There are a number of scalable implementations that can be used. Grid search plus cross-validation is one way to select the tuning parameter. According to our experiments, the procedure does not seem to be particularly sensitive to the choice of the tuning parameter. We will release our source code on GitHub and present the URL in the revised manuscript.

Computing the normalizing constant is a difficult problem even after the underlying structure model is inferred. When the dimension is low, it would be possible to estimate the normalizing constant via numerical integration, regardless of the kernel you are using.

Thank you for reading the paper in details and providing comments on how to improve the presentation.

Reviewer 4

Thank you for the detailed comments. We will carefully incorporate them in the revision.

Reviewer 5

Thank you for pointing out the relevant references. We will discuss about them in section Related Work. Our approach is similar to the references only in the sense that both use a nonparametric approach to estimate the undirected graph structure. However, our approach is substantially different from the references. First, our method is based upon score matching, which is then optimized by group lasso, while the references use the log-density ANOVA model, which requires computing the normalization constant once. Second, we provide a detailed proof about sparsistency property and convergence rate of our approach, while the references do not have such a result.

Reviewer 6

We will revise our manuscript to better explain our novelty and its relationship with other work.

Our established convergence rate is worse than that for a parametric model. However, our approach applies to a much larger class of models, not only the Gaussian/nonparanormal models. The method in [28] looks similar since it also uses score matching. However, the optimality conditions (prop 4.2 in our paper and Theorem 4 in [28]) of two objective functions are different, which leads to different algorithms and statistical analysis. We have cited all the relevant results in [28].

Reviewer 7

Thank you for your comments. We will significantly improve our manuscript to make it much easier to read.